# Coal Wall Spalling Mechanism and Grouting Reinforcement Technology of Large Mining Height Working Face

**DOI:** 10.3390/s22228675

**Published:** 2022-11-10

**Authors:** Hongtao Liu, Yang Chen, Zijun Han, Qinyu Liu, Zilong Luo, Wencong Cheng, Hongkai Zhang, Shizhu Qiu, Haozhu Wang

**Affiliations:** School of Energy and Mining Engineering, China University of Mining and Technology (Beijing), Beijing 100083, China

**Keywords:** roof subsidence, grouting reinforcement, roadway support, destruction of surrounding rock, dynamic pressure roadway

## Abstract

To control the problem of coal wall spalling in large mining height working faces subject to mining, considering the Duanwang Mine 150505 fully mechanized working face, the mechanism of coal wall spalling in working faces was investigated by theoretical analysis, numerical simulation and field experiment. Based on analysis of coal wall spalling in the working face, a new grouting material was developed. The stress and plastic zone changes affecting the coal wall, before and after grouting in the working face, were analyzed using numerical simulation and surrounding rock grouting reinforcement technology was proposed for application around the new grouting material. The results showed that: (1) serious spalling of the 150505 working face was caused by the large mining height, fault influence and low roof strength, and (2) the new nano-composite low temperature polymer materials used have characteristics of rapid reaction, low polymerization temperature, adjustable setting time, high strength and environmental protection. Based on analysis of the working face coal wall spalling problem, grouting reinforcement technology based on new materials was proposed. Industrial tests were carried out on the working face. Field monitoring showed that the stability of the working face coal wall was significantly enhanced and that rib spalling was significantly improved after comprehensive anti-rib-spalling grouting measures were adopted. These results provide a basis for rib spalling control of working faces under similar conditions.

## 1. Introduction

Comprehensive mechanized mining is often used in coal mining where the mining height of the fully mechanized mining face is significant, the strata pressure is relatively violent, and coal wall spalling is prone to occur [1,2]. Soft coal quality increases the probability of rib spalling. Following rib spalling, end-face roof caving can occur, which prejudices coal mine safety and affects the normal advancement of the working face, seriously restricting the efficient production of the mine [3,4].

The problem of coal wall spalling is related to the surrounding rock strength. Many researchers have investigated the strength of the rock mass through rock mechanics experiments, the results being of great importance for evaluation of the stability of the surrounding rock [5,6,7]. There are generally two major ways to prevent or reduce coal wall spalling [8,9,10,11]. First, the stress state of the coal wall of the working face can be improved, so that the unidirectional force of the coal body is avoided as far as possible to prevent stress concentration in the coal wall of the working face. This is mainly achieved by accelerating the advancing speed of the working face, timely beating of the guard panel, moving the frame with pressure, and increasing the initial support force of the support. Since the initial support force of the support cannot be infinitely increased, and the force provided by the guard panel is also limited, this first approach to preventing coal wall spalling is ineffective. A second approach is to alter the physical and mechanical properties of the coal, increasing the cohesion and strength of coal and ensuring the integrity of the coal. This approach is mainly effected through cement grouting and chemical grouting [12,13,14]. Therefore, it is important to identify low-temperature grouting materials which have good permeability, strong adhesion, low reaction heat, are of low cost, and can be widely applied in coal mines. Such materials are of great value for the prevention of coal wall spalling, improving coal production and increasing the economic benefits of coal-mining enterprises [15,16].

With respect to grouting support technology, based on problems associated with the large deformation of surrounding rock, serious fragmentation of the coal side and failure of the support body in a kilometer deep well, Kang Hongpu [17] proposed the use of high pressure anchor grouting-spraying synergistic control technology for a soft coal body and achieved good results in the field. Seeking to address the problem of large deformation of a deep coal seam, Xie Shengrong [18] proposed the use of external anchor-internal unloading collaborative control technology, which significantly improved the stress environment of the surrounding rock.

Using the 150505 working face of the Duanwang Mine as the engineering context, the coal wall spalling mechanism of a working face of significant mining height was studied through field investigation and numerical simulation. Based on analysis of the coal wall spalling problem, a new type of low-temperature polymer grouting reinforcement material was developed and use of surrounding rock grouting reinforcement technology suitable for the 150505 working face was proposed. The results of application of the proposed approach indicated that progress on the working face was enhanced, leading to economic benefits for the mine.

## 2. Engineering Background and Failure Characteristics of the Working Face

### 2.1. Engineering Overview

The coal seam of the 150505 fully mechanized mining face in the Wang Mine of the Shouyang Section of Shanxi Province is 15^#^ coal. The strike length of the working face is 288 m, the inclination length is 159 m and the mining area is 49,109.9 m^2^. The 15# coal type is a lean semi-bright-type coal, forming blocks and occurring where the coal seam is stable. The minimum thickness of the coal seam is 3.8 m, the maximum thickness is 4.25 m and the average thickness is 4.1 m. The minimum dip angle of the working face coal seam is 0°, the maximum dip angle is 15° and the average dip angle is 6°, so the occurrence of the working face coal seam is relatively stable. A plane layout of the working face and a histogram of the roof and floor of the rock are shown in Figure 1.

### 2.2. Overview of Coal Wall Spalling at the Working Face

The rib spalling problem has been a common problem in working faces with significant mining height, often leading to the support toppling and skewing, the scraper conveyor becoming stuck and the working face toppling, affecting the advancing speed of the working face. In severe cases, it may even cause significant casualties.

Mining the working face of a 15# coal seam is associated with coal seam occurrence stability and the thickness change is not substantial. The working face is a working face with large mining height. Since the roof of the coal seam is mainly sandy mudstone and the strength of mudstone is low, the support does not connect to the roof when the mining height is high and roof management is very difficult. When the support force is too large, the support is prone to mudstone collapse, spalling, roof fall and the occurrence of other accidents. Rib spalling of a coal wall in the process of working face advancement is shown in Figure 2.

## 3. New Polyurethane Composite Materials and Complete Technology for Coal and Rock Reinforcement

### 3.1. Development of New Grouting Materials

With respect to the causes of underground surrounding rock failure, the failure modes of the surrounding rock can be divided into two categories: failure induced by a structural plane or geological weak plane and failure induced by original rock stress. Soft rock failure and strength weakening are the fundamental causes of surrounding rock deformation. The surrounding rock shows different deformation characteristics at different stages. Therefore, strengthening the surrounding rock to improve the mechanical properties of the surrounding rock has become a key area of research. In light of the causes of weakening and failure of surrounding rock strength, grouting reinforcement technology applied to surrounding rock is strengthened in stages, and the corresponding point, line and surface reinforcements are carried out at different stages. The advantage of this approach is that it can improve the mechanical properties of the weak surface of the surrounding rock combined structure, enabling it to form a bearing structure and to ensure the self-stability of the surrounding rock. Moreover, the grout consolidation body can seal cracks and prevent the internal rock mass from soaking up water and gas, which is of great importance for maintaining the mechanical properties of the surrounding rock and ensuring its lasting stability [19].

Many types of chemical grouting materials have been developed with different grouting materials having different defects. Some materials have the disadvantages of polluting the environment, having a long cementation time, brittleness, irritating smell, flammability and of being harmful to the human body. As an ideal chemical grouting material, it should have the following characteristics [20]: good injectability, good durability, low reaction temperature, short curing time, good tensile and compressive strength, good permeability resistance, non-toxicity, odorlessness, low price and convenience of transportation. Up to now, most slurries do not fully meet the above requirements. In addition, in current applications of grouting materials, coal seam fire and smoke accidents have occurred in some mining areas during grouting. This is due to the high temperature produced by the rapid oxidation of the surrounding coal and rock during the chemical reaction of the grouting materials. Therefore, the grouting material must reduce the maximum reaction temperature of the coal mine reinforcement material grouting process.

Based on the characteristics of the above grouting materials, the new polyurethane composite material developed in this paper represents a new, low-temperature and safe liquid two-component environmentally friendly polymer material. It is produced by the mixing reaction of A and B two-component materials at low temperature with a volume ratio of 1:1. The main constituent of the A component is polyether polyol, while the main constituent of the B component is polymethylene polybasic polyisocyanate. The composition parameters are shown in Table 1.

The new material is produced as follows: First, polyether polyols and some isocyanate groups are reacted to generate prepolymers with high molecular weight with isocyanate groups at the end. Then, the prepolymers with isocyanate groups at the end are reacted with polyether polyol to form polycarbamate. The material has the following characteristics: when gelled, it is less affected by underground groundwater, has low reaction heat release, high adhesion, rapid strength increase, and has strong permeability resistance, abrasion resistance, stamping resistance and aging resistance. The grouting material is mainly suitable for the reinforcement of loose broken coal rock mass, roadway, small coal pillars and roadway in disrepair in the underground mining face. Advance grouting reinforcement to prevent top coal caving during excavation of reserved top coal is required.

The two-component materials are poured using a high-pressure grouting pump; the components A and B react quickly to form low exothermic and totally non-flammable elastomers, which are injected into the coal seam or rock strata by high pressure and can be extended to voids and cracks along the cracks of the soft coal and rock strata. The material can solidify in a short time and produce high strength cementitious materials to achieve the purpose of reinforcement; at the same time, the formed gel is subjected to secondary expansion and the loose coal seam in the non-grouting area is extruded so that the soft foundation coal seam forms a hard matrix to ensure the effective and safe mining of the shearer heading.

The structure of this material as shown under different high-resolution electron microscopes is shown in Figure 3.

To determine the physical and mechanical properties of the grouting materials, physical and mechanical tests were carried out in the laboratory. The standard test conditions in the laboratory were temperature 23 ± 2 °C and relative humidity 50 ± 5%. Before the tests, the samples were kept under standard test conditions for 24 h. The laboratory test diagram is shown in Figure 4 and the measured physical and mechanical parameters of the grouting materials are shown in Table 2.

### 3.2. Grouting Equipment and Grouting Technology

The supporting parts of the grouting equipment are shown in Figure 5. It is mainly composed of a grouting flower tube, hole packer, grouting tube, special injection gun, high pressure hose, grouting pump, A component material and B component material.

The grouting construction technology mainly includes: (1) the design of the hole position; (2) drilling; (3) installing the injection tube; (4) connecting the equipment air supply pipeline and grouting pipeline; (5) starting grouting; (6) stopping grouting; (7) replacement grouting; and (8) cleaning equipment and accessories after grouting. The detailed grouting process diagram is shown in Figure 6.

## 4. Numerical Simulation Verification of Grouting Effect

FLAC^3D^ 5.0 is a finite difference software package for studying geotechnical problems. In this paper, through numerical simulation, the stress and plastic zone changes of the coal wall before and after grouting were observed and compared, the peak position of the advanced abutment pressure of the coal body before and after grouting were determined and the failure range of the coal body in front of the coal wall were investigated.

### 4.1. Model Design Size

The model inclination length (X-direction) is 240 m, the strike length (Y-direction) is 200 m, and the height (Z-direction) is 67 m. According to the geological histogram of #15 coal in the Duanwang Mine, the model establishes 16 strata including the coal seam, a total of 3,245,547 nodes, and 3,168,000 grids with a minimum grid size of 1 m × 1 m × 1 m. The Mohr–Coulomb constitutive model is used in the model. Vertical stress is applied on the model boundary, which is calculated according to a depth of 230 m at the upper boundary of the model. The vertical stress is 6.5 MPa and the lateral pressure coefficient is 1.2. In addition to the upper boundary, the normal displacement constraint is applied on five boundaries. The numerical simulation model is shown in Figure 7. The mechanical parameter table of the surrounding rock mass is shown in Table 3.

The specific simulation steps are as follows: First, the vertical force is applied under the original rock stratum to simulate the pressure of the upper rock stratum. The upper boundary is a free constraint and the other boundary simulates the bottom layer with fixed constraints; the original stress is set horizontally. After the model is stable, the excavation simulation of the #15 coal is carried out. From x = 40 m and y = 75 m, the coordinate point, the working face with 160 m inclined length is excavated along the x-axis and the working face advances 50 m to the y-axis. Then, the model is calculated and, to be stable, the end and middle of the working face are sliced and the plastic zone and its stress nephogram before and after grouting are observed.

### 4.2. Comparative Analysis before and after Grouting

#### 4.2.1. Comparison of Advanced Abutment Pressure before and after Grouting

The distribution of the advanced abutment pressure before grouting is shown in Figure 8. It can be seen from Figure 8 that the advanced abutment pressure of the working face can be roughly divided into four regions:The stress reduction area is 0–3 m in front of the working face, which is the most significant area of coal fragmentation in front of the working face. The cracks in the coal body are crisscross and there is a risk that the whole coal body will fall off.Severely affected area, 3–10 m ahead of the working face. The area is the face of the advanced abutment pressure severely affected area. In this area, the stress value increases sharply to the peak stress, and then falls sharply. The stress on the coal within the scope of the region is the strongest with high stress causing the coal to be broken.The slowly decreasing area is 10–30 m in front of the working face. This area is the front part of the working face advanced abutment pressure influence area. The deformation increases slowly and the stress value decreases slowly, which has a continuous effect on the coal body in the area and the coal fracture is small in area.In the original rock stress area, beyond 30 m in front of the work, the stress value of coal in this area gradually decreases to the original rock stress and the integrity of coal in this area is good.

The advanced abutment pressure of the working face after grouting is shown in Figure 8. It can be seen from Figure 9 and Figure 10 that the advanced abutment pressure of the coal wall is consistent with the distribution before grouting, which first increases and then decreases. The coal and rock mass is relatively broken within 0–3 m and the abutment pressure is small. The stress concentration area is formed within 3–10 m. The maximum supporting pressure is 24.5 MPa, which is 0.5 MPa higher than that before grouting. The maximum peak point after grouting is about 3 m ahead of the working face and 2 m ahead of the peak point before grouting. From this point of view, grouting has obviously improved the strength and bearing capacity of the coal wall in the working face, strengthened the mechanical properties of the broken coal affected by mining, and improved the coal strength greatly. After grouting reinforcement, the cracks in the coal body are filled and the self-supporting and self-stability ability is strengthened. The failure conditions of the coal body are transformed from the original weak strength condition of the crack to the strength condition close to the coal body.

#### 4.2.2. Plastic Zone Analysis of Working Face before and after Grouting

Figure 11 is the plastic zone map of the working face before grouting. It can be seen from Figure 11 that the strength of the coal and rock mass is small; the failure range of the plastic zone in the coal wall of the working face is 2–5 m. The failure mode includes the ongoing tensile and shear failure that has occurred. The failure degree of the plastic zone in the roof rock is relatively complex and the failure range is highly covered to the upper rock. Near the left and right sides of the working face, the width of the plastic zone of the coal wall is small, being only 2 m. The working face tends to the middle position and the width of the plastic zone of the coal wall reaches 5 m. The width of the plastic zone of the coal wall is related to the advanced abutment pressure. The boundary of the plastic zone of the coal wall is the position where the advanced abutment pressure reaches a maximum value.

Figure 12 shows the distribution of the plastic zone at the end and middle of the coal wall of the working face after grouting. It can be seen from Figure 12 that, after grouting, due to the increase in coal strength at the coal wall of the working face, the failure width of the plastic zone of the coal wall is significantly reduced compared with that before grouting. The failure width of the plastic zone at the end is reduced from 2 m to 1 m and the failure width of the central plastic zone is reduced from 5 m to 3 m. The plastic failure region still includes ongoing tensile shear failure that has occurred. After the coal wall grouting, due to the reinforcement of the coal body, the stress concentration at the crack tip is weakened, the stress state of the rock mass near the crack is improved, and the strength and integrity of the coal body are improved, so that the overall bearing capacity of the coal wall of the working face is improved and the rib spalling of the working face is improved.

## 5. Grouting Scheme and Field Engineering Practice

### 5.1. Coal Wall Spalling Mechanism Analysis of Working Face

Based on investigation of the geological condition and numerical simulation of the 150505 fully mechanized mining face in the Duanwang Mine, the main causes of coal wall spalling in working face can be summarized as follows:

(1) Large Mining Height of Working Face

Since the working face is a fully mechanized working face with one-time mining, the working face has high mining height and high mining intensity, which makes the advanced abutment pressure of the working face larger and causes the coal body in the stress-increasing area to be broken. When the working face is pushed to the broken coal body position, the coal wall is easily broken under the extrusion of the support panel, resulting in a large area of rib caving. The probability and severity of coal wall spalling increase with mining height.

(2) Working Face Affected by Geological Structure

The geological exploration results of the geological structure in front of the working face show that there were nine faults in the working face, which influenced the continuous mining of the working face. When the working face passes the fault, the roof dislocation is greater and the support cannot provide sufficient support resistance to prevent roof subsidence. If the fault-affected area is large, the roadway roof cannot be effectively supported for a long time, which will generate significant security risks and affect the stability of the working face [21,22].

(3) Influence of Working Face Roof Lithology

The coal seam roof of the working face is composed of 1.31 m direct roof strata (limestone) and 11.47 m old roof strata (sandy mudstone). The strength of the mudstone is low, easily weathered in water and prone to collapse. Because the strength of the mudstone is low and it is easily broken, the support cannot effectively connect to the roof, which leads to coal wall spalling and roof caving.

(4) Influence of Roof Pressure on Coal Wall Spalling

Under the action of self-weight and roof pressure, the coal wall will show two failure modes of tensile fracture failure and shear failure. In the brittle and hard coal body, transverse tensile stress that cannot be alleviated or released by the deformation of the coal body is generated in the coal wall. When the tensile stress reaches the failure strength of the coal body, coal wall failure spalling occurs. The transverse deformation in the soft coal seam will relieve or release the tensile stress caused by compression; finally, shear sliding failure occurs because the shear stress in the coal wall is greater than the shear strength. In the large mining height working face, the mine pressure behavior is more intense. When the roof is near the initial pressure and periodic pressure, the abutment pressure of the coal wall of the working face reaches a peak value; the spalling is more likely to occur under the action of high roof pressure. After the rapid advance, the roof pressure is alleviated and the depth and length of the coal wall spalling will be reduced accordingly.

### 5.2. Grouting Scheme

In the working face, hole grouting is needed in the treatment stage. The working face is 4 m high, 2.5 m below the roof, 6 m deep, and 45° upward. In 150505, the return airway from the machine tail is 6 m, 2.5 m below the roof hole grouting, with a hole depth of 6 m, and an elevation angle 45° upward for the hole. The hole packer is placed 1–1.5 m away from the orifice. The pore distance preliminary design is 4 m with single hole grouting quantity control in 20 groups. A schematic diagram of the site grouting is shown in Figure 13. The diameter of the grouting borehole is 32 mm. To prevent the slurry from leaking in the roadway wall caused by excessive grouting pressure, the grouting pressure is 2–5 MPa.

### 5.3. Field Effect Analysis

By comparing the rib spalling of the working face before and after grouting, it was found that the number of rib spalling events in the coal wall after grouting was significantly reduced and that the average size of the rib spalling coal body was also significantly reduced, indicating that coal injection helped to prevent rib spalling of the coal wall. When the working face was pushed to the grouting section, the coal wall of the working face was observed and it was found that the coal wall formed a bonding layer; the grouting solution appeared around the obvious cracks, indicating that the slurry had created a bonding effect on the coal body. In addition, with advancement of the working face, the number and depth of coal wall spalling events in the working face were gradually reduced. During the final mining of the working face, the coal wall was basically flat and there was no coal wall spalling with large spalling depth. The grouting reinforcement effect was obvious and the efficient production of the working face was achieved while ensuring safety.

In addition, when the roof is broken, more than 10 people need to be assigned and production is stopped to maintain the roof. After application of this material, the construction is carried out during the maintenance shift, which does not affect the coal cutting of the production shift. Only three to four people are needed to organize the construction and labor productivity is increased by 150%. After grouting reinforcement, there is no longer a requirement to stop production to maintain the roof so that the coal mining work on the working face can occur continuously and as normal. On a monthly basis, over 1500 tons of raw coal mining were saved and raw coal production increased by 15%. The loss of coal due to the fault fracture zone was reduced and the recovery rate increased by 8.9%.

## 6. Conclusions

The main conclusions relating to coal wall spalling of the 150505 fully mechanized working face in Duanwang Mine are as follows:The working face has a large mining height, there are many faults and folds in the mining area and the roof of the coal seam is weak.A new nano-composite low temperature reinforcement material was developed, which has the characteristics of rapid reaction, low polymerization temperature, adjustable setting time, high strength and environmental protection.Through field observation and numerical simulation, the strength difference of the coal wall in the 150505 fully mechanized working face of Duanwang Mine, before and after grouting, was analyzed. It was found that the stability of the coal wall in the working face was significantly enhanced and rib spalling was significantly improved after comprehensive rib spalling prevention grouting measures were adopted.Based on investigation of the 150505 fully mechanized working face in the Duanwang Mine, a comprehensive prevention and control scheme of rib grouting reinforcement was proposed and industrial tests were carried out. The grouting reinforcement effect obtained was clear.

## Figures and Tables

**Figure 1 sensors-22-08675-f001:**
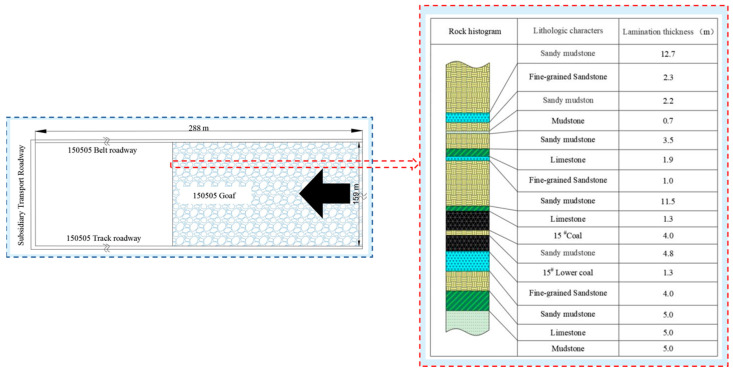
150505 working face layout relationship and comprehensive histogram.

**Figure 2 sensors-22-08675-f002:**
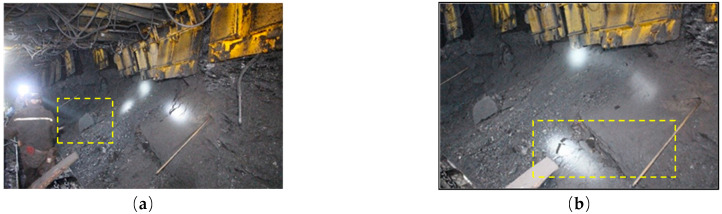
Failure diagram of working face. (**a**) Block collapse; (**b**) Support falling.

**Figure 3 sensors-22-08675-f003:**
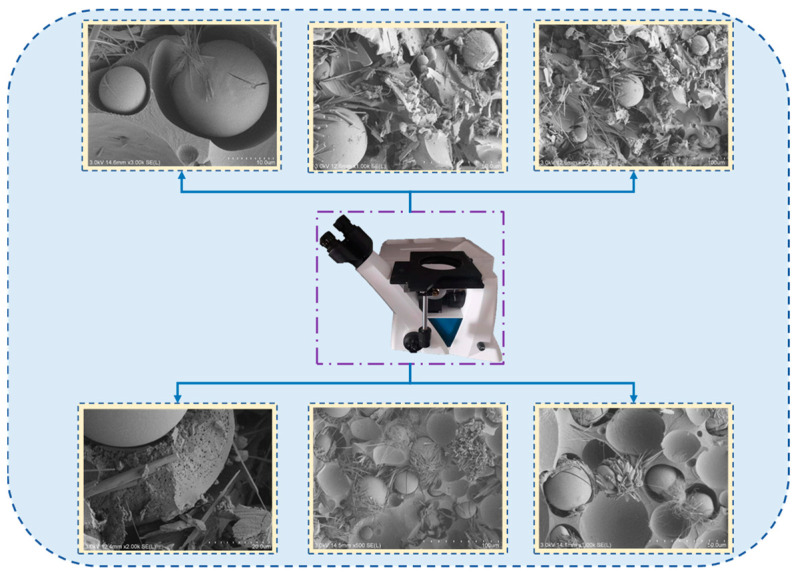
Images of new polyurethane composite materials under electron microscope.

**Figure 4 sensors-22-08675-f004:**
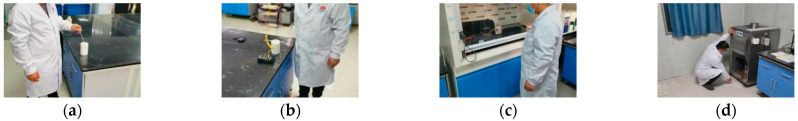
Laboratory test diagram. (**a**) Curing time test; (**b**) reaction temperature; (**c**) flashpoint test; (**d**) flame-retardant performance test.

**Figure 5 sensors-22-08675-f005:**
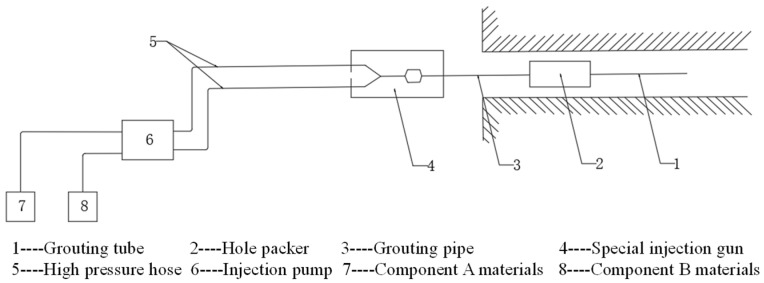
Part drawings of grouting equipment.

**Figure 6 sensors-22-08675-f006:**
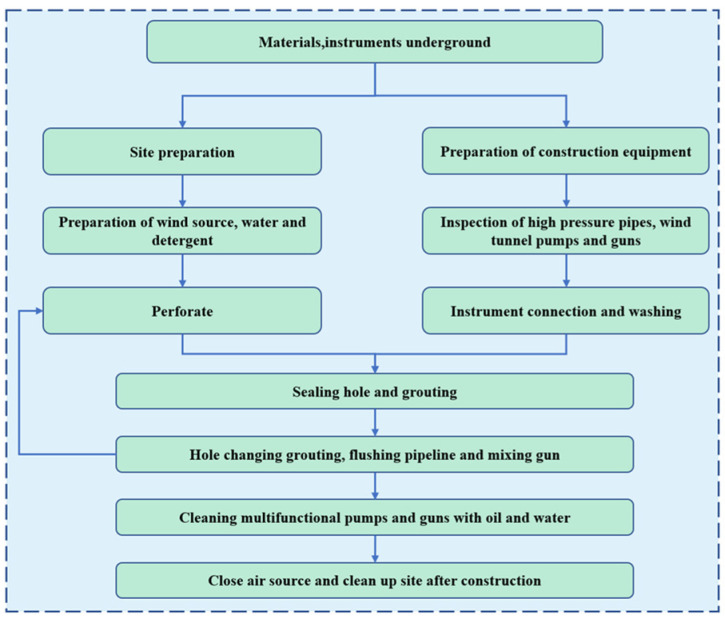
Grouting process flowchart.

**Figure 7 sensors-22-08675-f007:**
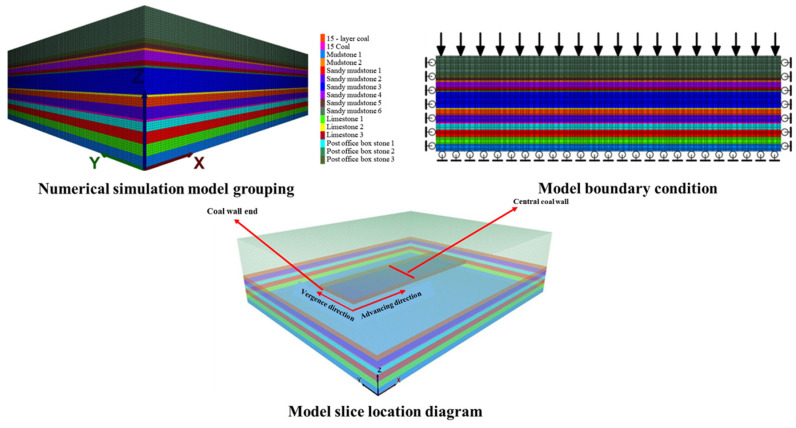
Numerical simulation model diagram.

**Figure 8 sensors-22-08675-f008:**
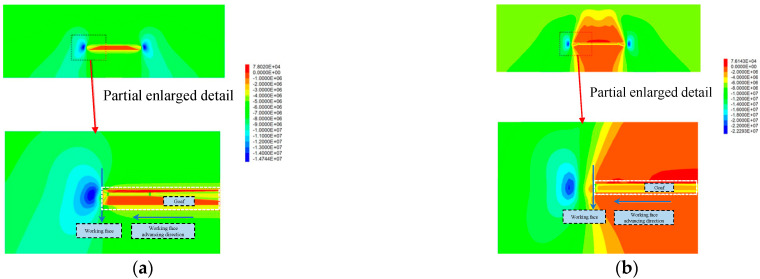
Local stress nephogram before grouting. (**a**) End section; (**b**) Middle section.

**Figure 9 sensors-22-08675-f009:**
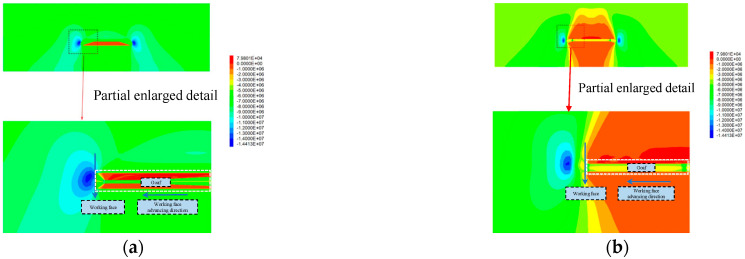
Local stress nephogram after grouting. (**a**) End section; (**b**) Middle section.

**Figure 10 sensors-22-08675-f010:**
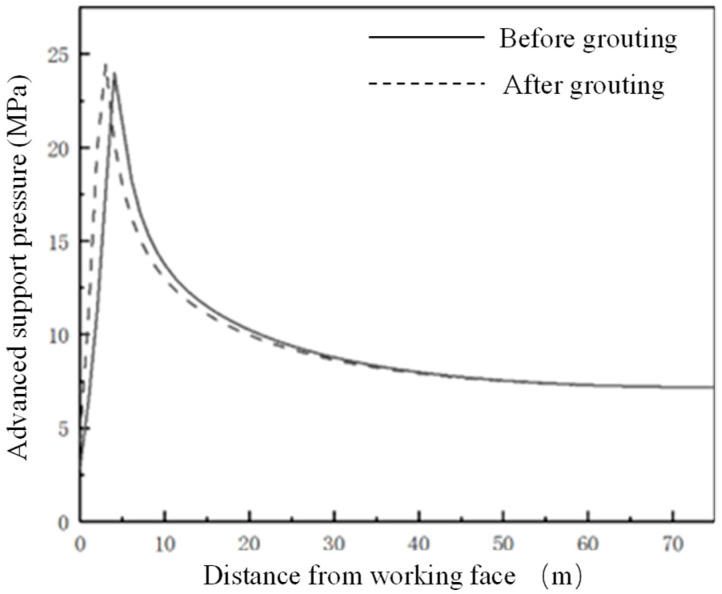
Comparison of advanced abutment pressure before and after grouting.

**Figure 11 sensors-22-08675-f011:**
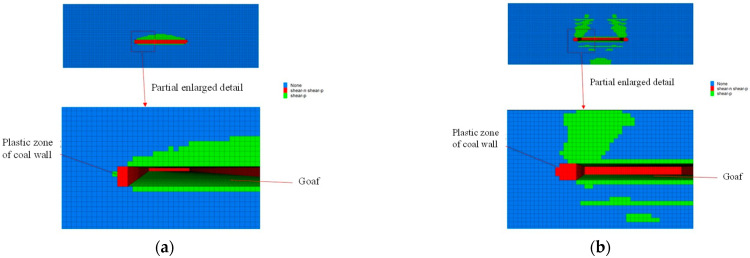
Plastic zone of coal wall before grouting. (**a**) Plastic zone of end coal wall; (**b**) Plastic zone of central coal wall.

**Figure 12 sensors-22-08675-f012:**
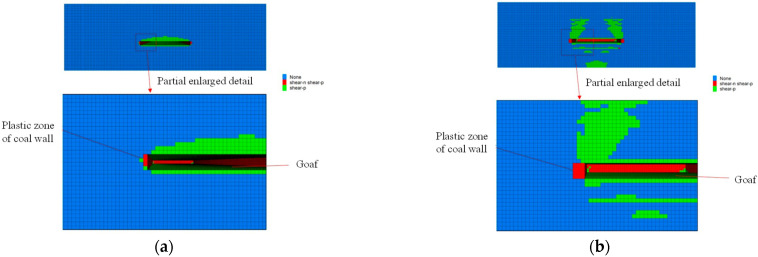
Plastic zone of coal wall after grouting. (**a**) Plastic zone of end coal wall; (**b**) Plastic zone of central coal wall.

**Figure 13 sensors-22-08675-f013:**
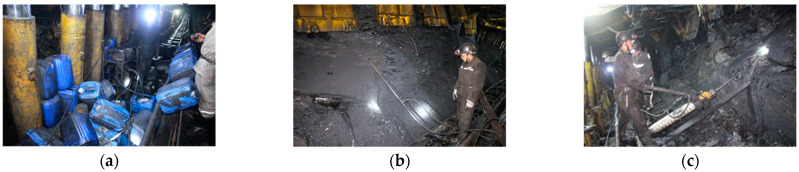
On-site grouting schematic diagram. (**a**) Grouting pump and connecting injection gun; (**b**) Connection between grouting pipe and grouting gun; (**c**) Perforate.

**Table 1 sensors-22-08675-t001:** Material composition table.

Component A:	Weight/%	Component B:	Weight/%
Raw Material Name		Raw Material Name	
Polyol	0–98	Isocyanate	65–100
Foam over stabilizer	0–2.0	Polyol	0–10
Catalyst	0.4–0.8	Plasticizer	0–35

**Table 2 sensors-22-08675-t002:** Technical index of grouting material.

Project		Standard Requirements	Test Results	
			Component A	Component B
Appearance		Homogeneity, No caking	Homogeneity; No caking	
Flashpoint /°C		≥100	225	Not detected
Curing time (23 ± 2) °C/s		——	40~100	
Expansion multiple /times		≥1.0	2.0~3.0	
Ant aging performance (80 °C, 168 h)		No surface change, No mass loss	No surface change; No mass loss	
Maximum reaction temperature /°C		≤140	80~90	
Bond strength /MPa		≥3.5	4.2	
Flame-retardant and anti-static properties		Meet standards MT113	Non-combustible; anti-static performance meets the requirements of standard MT113	
Harmful	Benzene /g/kg	≤5.0	Not detected	
Material	Toluene + Xylene /g/kg	≤150	Not detected	
Limit	Total volatile organic compounds /g/L	≤700	44	
Quality guarantee period (At room temperature/month)		——	≥6	≥6

**Table 3 sensors-22-08675-t003:** Rock mechanics parameters.

Serial Number	Lithologic Characters	Lamination Thickness (m)	Total Thickness (m)	Bulk Modulus (GPa)	Shear Modulus (GPa)	Cohesion (MPa)	Angle of Internal Friction (°)	Tensile Strength (MPa)
16	sandy mudstone	12.70	67.20	7.01	4.50	1.93	29	2.50
15	fine-grained sandstone	2.30	54.50	8.95	5.89	2.12	29	3.00
14	sandy mudstone	2.20	52.20	7.01	4.50	1.93	29	2.50
13	mudstone	0.70	50.00	4.01	3.50	1.50	28	2.00
12	sandy mudstone	3.50	48.30	7.01	4.50	1.93	29	2.50
11	limestone	1.90	44.80	11.95	8.87	2.51	32	4.0
10	fine-grained sandstone	1.00	42.90	8.95	5.89	2.12	29	3.00
9	sandy mudstone	11.50	41.90	7.01	4.50	1.93	29	2.50
8	limestone	1.30	20.40	11.95	8.87	2.51	32	4.00
7	#15 coal	4.00	29.10	3.98	2.50	1.03	22	1.50
6	sandy mudstone	4.80	25.10	7.01	4.50	1.93	29	2.50
5	#15 lower coal	1.30	20.30	3.98	2.50	1.03	22	1.50
4	fine-grained sandstone	4.00	19.00	8.95	5.89	2.12	29	3.00
3	sandy mudstone	5.00	15.00	7.01	4.50	1.93	29	2.50
2	limestone	5.00	10.00	11.95	8.87	2.51	32	4.00
1	mudstone	5.00	5.00	4.01	3.50	1.50	28	2.00

## Data Availability

Not applicable.

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
