# Peer review of "Coal Wall Spalling Mechanism and Grouting Reinforcement Technology of Large Mining Height Working Face"

_sensors, 2022, doi:10.3390/s22228675_

Round 1
Reviewer 1 Report
Manuscript sensors-1937196 investigates the mechanism of coal wall spalling in large mining height working face through theoretical analysis, numerical simulation and field test, and propose a solution to the spalling problem in 150505 working face of the Duanwang Mine based on the grouting materials developed. The manuscript is novel and its content falls into the scope of Sensors. However, here are some of the minor revision before published. The comments are as follows:
1. There are several published papers solving the problems at large mining height working face, literature review of these papers would assist with coming up with the innovation of this manuscript. Selected suggested papers as list as follows, however, the literature review is not limited to the listed papers.
(1)Zhu Sitao, Feng Yu, Jiang Fuxing, et al. Mechanism and risk assessment of overall-instability-induced rockbursts in deep island longwall panels, International Journal of Rock Mechanics and Mining Sciences[J]. 2018, 106(6): 342-349.
(2)Zhu Sitao, Feng Yu, Jiang Fuxing. Determination of abutment pressure in coal mines with extremely thick alluvium stratum: a typical kind of rockburst mines in China[J]. Rock Mechanics and Rock Engineering, 2016, 49(5): 1943-1952.
(3)Wang Bo, Zhu Sitao, Jiang Fuxing, et al. Investigating the Width of Isolated Coal Pillars in Deep Hard-Strata Mines for Prevention of Mine Seismicity and Rockburst[J]. Energies, 2020, 13(17):4293.
2. Page 2. Rows 52-56 indicate that the price is also an important consideration for the ideal grouting material. Is the new type of low-temperature polymer grouting reinforcement material more expensive than conventional grouting products? In Figure 1, 15 coal and the above should be rewritten as 15# coal.
3. Page 3, line 90. The title number repeats the above, similarly, the following also needs to be modified. Does the newly developed grouting material have relevant laboratory physical and mechanical performance tests? What are the test parameters?
4. Page 6. The text in Figure 6 is partially obscured, and there should be a lithology mark on the legend side, or it can be omitted. The retained decimal digits of each parameter in Table 2 should be unified.
5. Page 7, Line 200-201. “The internal cracks of the coal are crisscrossed horizontally, and even the whole rib spalling and falling off of the coal will occur.”. Statement logic is not clear, horizontally crisscross refers to what?
6. Page 10. Article 4.3 site effect analysis part of the new nano composite low temperature reinforcement material in the field application effect is good, whether there are specific data to quantify the specific grouting effect? At the same time, comparative analysis can also be considered in combination with on-site working condition pictures before and after grouting.
7. Page 11. It is concluded that the mechanism of coal wall spalling in 150505 fully mechanized working face of Duanwang Mine is related to many faults and folds in the mining area, but there is no such description in this manuscript.
Author Response
Dear Reviewers:
Thank you for your letter and for the reviewers’ comments concerning our manuscript,We appreciate for Reviewers' warm work earnestly. We will send the revised manuscript and cover letter to you through the attachment, Thanks very much for your attention to our paper.
Very sincerely yours,
Yang Chen

Reviewer 2 Report
The manuscript "Coal wall spalling mechanism and grouting reinforcement technology of large mining height working face" obtained Coal wall spalling mechanism through simulation test and theoretical analysis. At the same time, this manuscript also developed a new type of grouting reinforcement material, which provided a guarantee for safe production. However, there are still the following deficiencies in this manuscript. Authors are requested to answer in detail before being published. The opinions are as follows :
1. The 150505 working face appears many times in the abstract. Since the full text only describes and analyzes this working face, it is suggested to simplify the subsequent 150505 working face in the abstract to this working face.
2. According to the text described in the development of new grouting materials have excellent performance, but the text does not have specific physical and mechanical parameters of grouting materials, or the lack of performance with other grouting materials contrast ?
3. Section 2.2 and field test part of the lack of such as grouting pressure, aperture grouting parameters? Authors may supplement the applicable scope and conditions of the new grouting material underground.
4. In Section 3.1, 6.5Mpa should be MPa, please check the similar error. Some pictures in the text are not fully displayed in Figure 7, please modify and check similar errors.
5. Grouting support as a common means of support, authors should thinks what should be paid attention to in the process of grouting reinforcement ?
6. Does the author consider the influence of roof pressure on coal wall spalling in the analysis of rib spalling mechanism ?
7. The authors are advised to add some important references about“grouting reinforcement technology”.
Author Response

(The authors gave the same response as above.)
